



# Can statistics of turbulent tracer dispersion be inferred from camera observations of $SO_2$ in the ultraviolet?

Arve Kylling[1], Hamidreza Ardeshiri[1], Massimo Cassiani[1], Anna Solvejg Dinger[1], Soon-Young Park[2], Ignacio Pisso[1], Norbert Schmidbauer[1], Kerstin Stebel[1], and Andreas Stohl[1]

[1]NILU - Norwegian Institute for Air Research, NO-2007 Kjeller, Norway
[2]Center for Earth and Environmental Modeling Studies, Gwangju Institute of Science and Technology, Gwangju, Republic of Korea

*Correspondence to:* Arve Kylling (arve.kylling@nilu.no)

**Abstract.** Turbulence is one of the unsolved problems of physics. Atmospheric turbulence and in particular its effect on tracer dispersion may be measured by cameras sensitive to the absorption of ultraviolet (UV) sun-light by sulfur dioxide ($SO_2$), a gas that can be considered a passive tracer over short transport distances. We present a method to simulate UV camera measurements of $SO_2$ with a 3D Monte Carlo radiative transfer model which takes input from a large eddy simulation (LES)

5  of a $SO_2$ plume released from a point source. From the simulated images the apparent absorbance and various plume density statistics (centerline position, meandering, absolute and relative dispersion, skewness, and fractal dimension) were calculated. These were compared with corresponding quantities obtained directly from the LES. Mean differences of centerline position, absolute and relative dispersion, and skewness between the simulated images and the LES were found to be smaller than a quarter of one camera pixel, with standard deviations between 1/2 and 3/2 camera pixel. Furthermore sensitivity studies were

10  made to quantify how changes in solar azimuth and zenith angles, aerosol loading (background and in plume), and surface albedo impact the UV camera image plume statistics. Changing the values of these parameters within realistic limits have negligible effect on the centerline position, meandering, absolute and relative dispersions, and skewness of the $SO_2$ plume. Thus, we demonstrate that UV camera images of $SO_2$ plumes may be used to derive plume statistics of relevance for the study of atmospheric turbulent dispersion.



# 1 Introduction

Air motion in the lowest part of the atmosphere is over land bounded by a solid surface of varying temperature and roughness. This part of the atmosphere is named the planetary boundary layer (PBL) (Stull, 1988). It responds quickly to surface radiation changes, and the air motion in the PBL is nearly always turbulent. A substance released into this turbulent atmosphere will experience concentration fluctuations at locations downwind of its source that are important, particularly if responses are non-linear; for example with respect to toxicity, flammability and odour detection (e.g. Hilderman et al., 1999; Schauberger et al., 2012; Gant and Kelsey, 2012) and non-linear chemical reactions (Brown and Bilger, 1996; Vilà-Guerau de Arellano et al., 2004; Cassiani et al., 2013). The complete description of turbulence remains one of the unsolved problems of physics. The COMTESSA project (Camera Observation and Modelling of 4D Tracer Dispersion in the Atmosphere; https://comtessa-turbulence.net/) aims to "elevate the theory and simulation of turbulent tracer dispersion in the atmosphere to a new level by performing completely novel high-resolution 4D measurements". To achieve this, six UV cameras have been built to measure $SO_2$ densities from various viewing directions. A series of experiments with puff and continuous releases of $SO_2$ from a tower have been performed as described by Dinger et al. (2018). It is known from measurements of volcanic $SO_2$ emissions that aerosol and viewing geometry affect the retrieved $SO_2$ amounts (Kern et al., 2013). Furthermore, variations in surface albedo and solar zenith and azimuth angles may have an impact. The influence of these factors on the UV camera images, the deduced $SO_2$ amounts and density statistics needs to be quantified and, if necessary, corrected for.

The aims of this paper are to present a novel method to simulate UV camera images of a dispersing $SO_2$ plume based on a large eddy simulation (LES) of such a plume and to examine how various factors (solar angles, aerosol content, and surface albedo) affect the statistical parameters characterizing the $SO_2$ plume dispersion. The large eddy simulation (LES) providing the input to the radiative transfer modelling, the radiative transfer model used to simulate the camera images and the statistical parameters are described in section 2. The effects of solar azimuth and zenith angles, surface albedo, background aerosol, and aerosols in the plume on plume density statistics are presented in section 3. Furthermore, the plume density statistics from the simulated images are compared with statistics derived directly from the LES simulations. The paper ends with the conclusions in section 4.

# 2 Methods

## 2.1 Large eddy simulation (LES)

Large eddy simulation is nowadays viewed as a popular tool in many applied atmospheric dispersion studies, especially of the urban environment and for critical applications such as the release of toxic gas substances (e.g. Fossum et al., 2012; Lateb et al., 2016). LES provides access to the three dimensional turbulent flow field and it is sometimes used as a replacement for experimental measurements at high Reynolds numbers. In this methodology, the large scales of the turbulent flow are explicitly simulated while a low-pass filter is applied to the governing equations to remove the small scales information from the numerical solution. The effects of the small scales are then parameterized by means of a sub-grid scale (SGS) model





(e.g. Deardorff, 1973; Moeng, 1984; Pope, 2000; Celik et al., 2009). We used the Parallelized Large-Eddy Simulation Model (PALM, Raasch and Schröter, 2001; Maronga et al., 2015) to solve the filtered, incompressible Navier-Stokes equations in Boussinesq-approximated form, at infinite Reynolds number. A three dimensional domain of $1000\,\text{m} \times 375\,\text{m} \times 250\,\text{m}$ in the along wind ($x$), crosswind ($y$) and vertical ($z$) directions respectively, was simulated with a grid resolution of $nx \times ny \times nz = 1024 \times 384 \times 256$. Here $nx, ny, nz$ are the number of grid nodes in along wind, crosswind and vertical directions, respectively. This implies that the size of a grid cell is $0.98^3\,\text{m}^3 \approx 1\,\text{m}^3$. The release point was located at $25\,\text{m}$ above the ground.

The neutral boundary layer was simulated as an incompressible half channel flow at an infinite Reynolds number. The flow was driven by a constant pressure gradient. For the velocity, periodic boundary conditions were used on the lateral boundaries while on the top, strictly symmetric, stress free, condition was applied. The bottom wall was not explicitly resolved but a constant flux layer was used as is commonly done in atmospheric simulations. Non-periodic boundary conditions were set for the passive scalar. For further information on the model set-up see also Ardeshiri et al. (2019).

The LES calculates 3D $SO_2$ concentrations as a function of time. The $SO_2$ concentrations are used as input to the 3D radiative transfer model simulations. A total of 100 time frames were calculated with a time resolution of 6.25 s. For the sensitivity studies one randomly chosen time frame was used, while seven randomly chosen frames were used for the reference case and the fractal dimension calculation (see section 3). $SO_2$ column densities for one instant of the LES simulation are shown along the y and z-axes in Figs. 1a and 1b, respectively, for the part of the plume viewed by the camera.

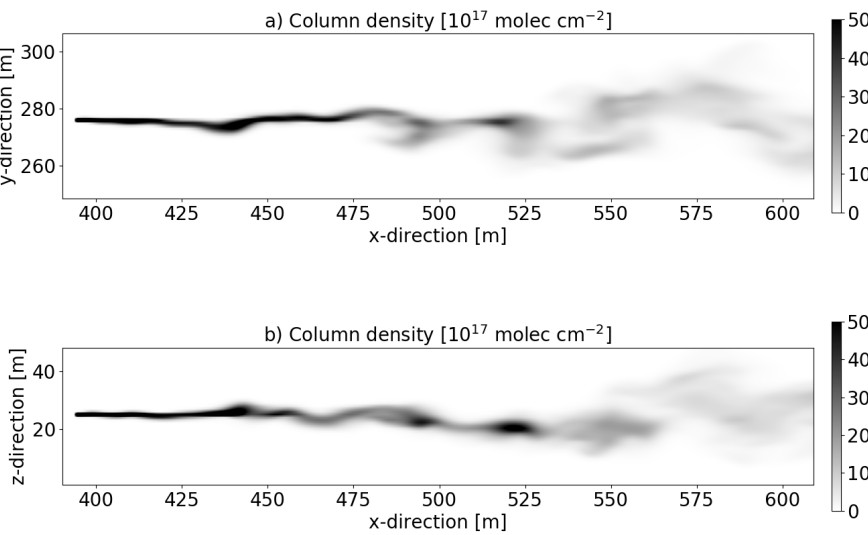

**Figure 1.** a) $SO_2$ column densities from the LES (time frame no. 5) integrated along the vertical z-direction. b) $SO_2$ column densities integrated along the cross-wind y-direction.



## 2.2 Radiative transfer simulations

The UV camera images were simulated with the 3D MYSTIC Monte Carlo radiative transfer model which was run within the libRadtran framework (Mayer et al., 2010; Emde et al., 2010; Buras and Mayer, 2011; Mayer and Kylling, 2005; Emde et al., 2016). MYSTIC includes an option to calculate the radiation impinging on a camera with a prescribed number of pixels in a plane defined by the location of the camera within a 3D domain and the camera viewing direction. For this option the MYSTIC Monte Carlo model is run in backward mode. The MYSTIC camera simulation capabilities have earlier been used by for example Kylling et al. (2013) to simulate infrared satellite images. Here it is used to simulate radiative transfer to the UV camera at wavelengths suitable for the detection of $SO_2$. Thus, for each camera pixel, spectra were calculated for wavelengths ranging between 300 and 350.5 nm. The spectral resolution was 0.1 nm in order to capture the fine structure of the $SO_2$ cross section. The spectra were weighted with spectral response functions (about 10 nm width) representing cameras with mounted on-band (sensitive to $SO_2$ absorption, centred at 310 nm) and off-band (barely sensitive to $SO_2$ absorption, centred about 330 nm) filters similar to those described by Gliß et al. (2018). Quantum efficiency of the detector and geometrical effects related to lens/camera optics are not included in the camera simulations. $SO_2$ plume concentrations were adopted from the LES simulations described in section 2.1 and the spectrally dependent $SO_2$ absorption cross section was taken from Hermans et al. (2009).

A finite 3D domain (bird's eye view provided in Fig. 2) is defined for the radiative transfer simulations. The $SO_2$ plume is embedded in this domain and is viewed from the side at a distance of about 250 m by the UV camera which is placed 1 meter above the surface. This camera-plume distance is comparable to that used during part of the first COMTESSA field campaign described by Dinger et al. (2018).

As the COMTESSA field campaigns are being carried out primarily in central Norway during the summer time, solar zenith angles of 40° and 60° were considered. When not otherwise noted (see section 3.2), the sun was assumed to be perpendicular to the camera viewing direction, see Fig. 2.

The LES voxel resolution is about 1 m³ which at a distance of 250 m corresponds to 0.004 rad=0.23°. The simulated camera field of view (FOV) was set to 46°×10° to cover the main plume. To ensure sufficient spatial sampling the camera resolution was specified to be about half the LES voxel resolution. With the FOV and a camera resolution of about 0.5 m at the plume, this gives a camera with 400×88 pixels. It is noted that the UV cameras used by Dinger et al. (2018) had 1392×1040 pixels. The reason and justification for using fewer pixels in the simulated camera are twofold: 1) With the simulated camera it is possible to zoom onto the plume as one always knows where the plume is. In an experimental setting, the plume usually covers only part of the FOV, to allow for changes in wind direction and, thus, changes in plume position; 2) The computer time and memory requirements increase as the number of pixels increases. It is thus advantageous to use as few pixels as needed to cover the plume.

To further save computer memory and time, a full 3D description of the plume is given only in the part of the domain containing the plume seen by the camera (red square in Fig. 2). Outside the red square, the plume is not included. Energy conservation is ensured by using circular boundary conditions, that is, photons leaving the domain on one side enter the domain



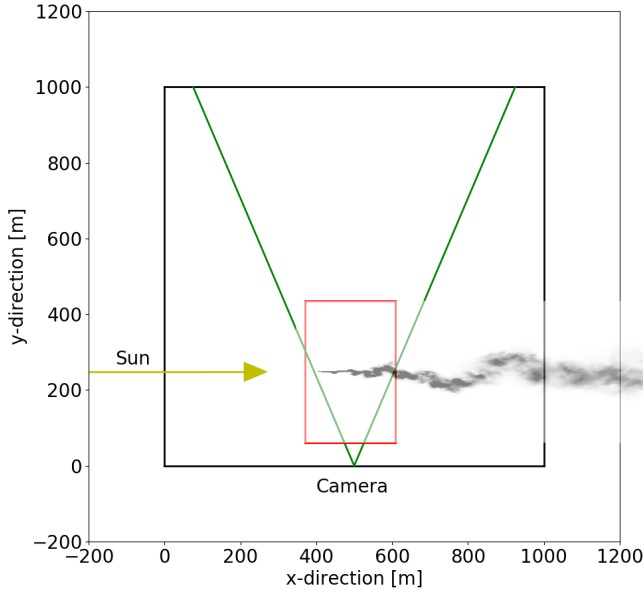

**Figure 2.** Bird's eyes view of the 3D domain (black square) and the SO$_2$ plume location within the domain (red square). The UV camera is located where the two green lines intersect. They indicate the camera's horizontal field-of-view. The column density (similar to the upper plot in Fig. 1) of the plume is included for illustrative purpose. The direction of the incoming Sun ray is shown by the yellow line.

again on the opposite side. This implies that the plume may be seen several times by the camera if not enough care is taken when setting up the geometry of the computational domain, the location of the plume within the domain, and the camera. This is challenging in the present situation where a realistic experimental setup is simulated (low altitude plume, camera viewing angles close to the horizon). Great care was thus taken when setting the domain size and the camera field of view to avoid

"ghost" plumes in the simulated images. Still, part of a secondary "ghost" plume is present in some of the images. These "ghost" plumes have been removed from the analysis presented below.

For representing the ambient atmosphere the mid-latitude summer atmosphere of Anderson et al. (1986) was used. The surface albedo is small in the UV for non-snow covered surfaces and was thus set to zero when not otherwise noted (see section 3.4). Aerosols were included for specific sensitivity tests that are described in section 3.3.

The radiative transfer simulations were run on a Linux cluster utilizing 10 processors in parallel with each process needing about 10-15 GB of memory depending on whether aerosols in the plume were included or not. The MYSTIC Monte Carlo radiative transfer simulation is statistical in nature and the simulated images thus contain statistical noise. To achieve a noise level of about the same order of magnitude as the measurements ($\approx 1\%$), a sufficient number of photons needs to be traced. For each pixel and wavelength 2000 photons were traced. This gave simulation times for one on- and one off-band image of

about 120-140 hours, and ensured that for the simulations without aerosol and zero surface albedo at least 93.0% of the pixels



had radiances with a relative standard deviation $<1.0\%$. For simulations with background aerosols the corresponding number is 83%.

## 2.3 Analysis methodology

The apparent absorbance for the on-band camera is given by Mori and Burton (2006) and Lübcke et al. (2013)

$$\tau_{on} = -\ln \frac{I_{on,M}}{I_{on,0}} \tag{1}$$

Here, $I_{on,M}$ is the on-band radiance and $I_{on,0}$ the background radiance without the $SO_2$ plume. In addition to absorption by $SO_2$, $\tau_{on}$ may include absorption due to aerosol and plume condensation. Assuming that the absorption by these other constituents varies little with wavelength between the on- and off-band cameras, the extra absorption may be removed by subtracting the off-band absorption:

$$\tau = \tau_{on} - \tau_{off} = -\ln \frac{I_{on,M}}{I_{on,0}} + \ln \frac{I_{off,M}}{I_{off,0}} = \ln \left( \frac{I_{off,M}}{I_{on,M}} \frac{I_{on,0}}{I_{off,0}} \right), \tag{2}$$

where $I_{off,M}$ and $I_{off,0}$ are the off-band radiance and the off-band background radiance respectively. The background images were calculated similar to the plume images, but with the $SO_2$ concentration set to zero. Below, plume statistics are presented for both $\tau_{on}$ and $\tau$.

Ideally, plume statistics from the LES $SO_2$ concentrations and image derived $SO_2$ concentrations should be compared. However, for the images this would require simulating a geometry suitable for tomography and tomographic reconstruction of the plume. The slant column density (SCD) is the concentration of a gas along the light path (typically in units of m$^{-2}$). It may be readily calculated from the LES $SO_2$ concentrations. From apparent absorbances the SCD may be retrieved. For $SO_2$ camera measurement this is done by calibrating the camera with $SO_2$ cells and/or concurrent differential optical absorption spectroscopy (DOAS) measurements. The calibration gives a linear relationship between the apparent absorbance and the SCD (see for example Lübcke et al., 2013). Such calibration procedures could be simulated and used to calibrate the simulated images. However, higher order moments (first order moment and upwards) would be the same for the SCD and the apparent absorbance due to the linear relationship between the two. Thus, below we will compare $SO_2$ SCD from the LES with apparent absorbance from the images. We note that this comparison will not include the zeroth moment (total mass) and that systematic biases may go undetected. While a comparison of the total mass certainly is of interest, this would require a systematic investigation of $SO_2$ calibration using simulated images, which is beyond the scope of this work.

### 2.3.1 Plume statistics

For projected LES simulations and the simulated images, the vertical (in the images) plume centerline position, meandering, absolute and relative dispersions, and the skewness were calculated (see e.g. Dosio and de Arellano, 2006). Each pixel in the camera images and each projection from the LES simulations describe the integrated column amount ($\rho_L$) of the trace gas along the line of sight ($dL$):

$$\rho_L = \int \rho dL \tag{3}$$





where $\rho$ is the density of the trace gas. The mean plume height $\overline{z}$ and the instantaneous vertical plume centerline position $z_m$ are given by

$$\overline{z} = \frac{\iint z\rho_L dx dz}{\iint \rho_L dx dz} \tag{4}$$

$$z_m(x) = \frac{\int z\rho_L dz}{\int \rho_L dz} \tag{5}$$

where $z$ is the pixel number in the vertical direction and $x$ is the pixel number in the horizontal direction. We note that strictly speaking world coordinates should be used instead of pixel numbers in the equations above and below. However, for the geometry in this study the difference between the two is less than 1%. Thus, for convenience we use pixel numbers.

Following Dosio and de Arellano (2006) we define the fluctuations of the absolute, relative and centerline positions:

$$z' = z - \overline{z} \tag{6}$$

$$z_r = z - z_m \tag{7}$$

$$z'_m = z_m - \overline{z}. \tag{8}$$

Ideally, some time-averaged value of $\overline{(z)}$ should be used in Eqs. 6-8. Here we calculate $\overline{(z)}$ from a single time step. This implies that there will be correlations between $z'_m$ at different $x$, which also implies that $\overline{(z'_m)}$ is strictly speaking not a robust reference for defining meandering of the plume. However, the purpose of this study is to investigate the sensitivity of the statistical

properties to changes in various atmospheric parameters and this limitation should have minimal impact on the results.

The absolute ($\sigma_z$), relative ($\sigma_{zr}$) and meandering ($\sigma_{zm}$) dispersions are defined as:

$$\sigma_z^2(x) = \frac{\int \rho_L z'^2 dz}{\int \rho_L dz} \tag{9}$$

$$\sigma_{zr}^2(x) = \frac{\int \rho_L z_r^2 dz}{\int \rho_L dz} \tag{10}$$

$$\sigma_{zm}^2 = \frac{\int \rho_L z'^2_m dx}{\int \rho_L dx} \tag{11}$$

and similarly for the skewnesses:

$$\overline{z'^3} = \frac{1}{\sigma_z^3(x)} \frac{\int \rho_L z'^3 dz}{\int \rho_L dz} \tag{12}$$

$$\overline{z'^3_{zr}} = \frac{1}{\sigma_{zr}^3(x)} \frac{\int \rho_L z_r^3 dz}{\int \rho_L dz} \tag{13}$$

$$\overline{z'^3_{zm}} = \frac{1}{\sigma_{zr}^3(x)} \frac{\int \rho_L z'^3_m dx}{\int \rho_L dx} \tag{14}$$

These various quantities were calculated both directly from the projected LES simulations and also from the camera images.

The former served as a reference ("ground truth") against which the quantities derived from the camera images were compared.



### 2.3.2 Fractal dimension

Turbulent flow is non-uniform. Thus, besides describing the flow with various moments as explained in section 2.3.1, it may be fruitful to use fractal methods to analyse tracer dispersion. One method for determining the fractal dimension is the box-counting algorithm. The box counting dimension, $D_{box}$, or Minkowski-Bouligand dimension, is defined as (see for example Feder, 1988)

$$D_{box} := \lim_{\epsilon \to 0} \frac{\log(N(\epsilon))}{\log(1/\epsilon)}. \tag{15}$$

Here $\epsilon$ is the side length of the boxes used to measure the number of boxes $N(\epsilon)$ covering the feature of interest. Here we define the feature of interest to be the $SO_2$ plume where the apparent absorbance $\tau > 0.03$, see Eq. 2 and Fig. 3. The threshold on $\tau$ is set in order to avoid influence from noisy pixels. From the threshold filtered and gray-scaled $\tau$-images the fractal dimension was estimated using the mass box counting method. This was done both for projected LES simulations and simulated UV camera images. The Fractal Dimension and Lacunarity (FracLac) plugin to the Image processing and analysis in Java (ImageJ) software (Karperien, 1999-2013; Rasband, 1997-2018), was used to calculate the fractal dimension.

## 3 Results

We first compare statistical results from the LES and simulated images reference atmospheric conditions. This comparison is made for both single and multiple time frames and is done in order to estimate how well the camera derived statistics may reproduce the LES statistics. Next the impact of solar angles, aerosol load, and surface albedo on plume statistics are investigated for a single time frame.

### 3.1 Reference cases

Figs. 3a and 3b show simulated on- and off-band radiances, respectively, for the reference case (no aerosol, zero surface albedo and a solar zenith angle of $40°$). The strong absorption of $SO_2$ in the on-band image is clearly visible in Fig. 3a. There is also a weak $SO_2$ signal in the off-band image (Fig. 3b). From the on- and off-band images, and corresponding background images not including the $SO_2$ plume, the apparent absorbance was calculated using Eq. 2. The resulting apparent absorbance is shown in Fig. 3c on a linear scale and in Fig. 3d on a logarithmic scale.

The plume centerline, absolute and relative dispersion, and skewness, as defined in Eqs. 4, 9 and 12, were calculated directly from the LES density values in Fig. 1. These are shown as solid lines in Fig. 4. The same quantities were derived from the apparent absorbance in Fig. 3 and are shown as dotted lines in Fig. 4 for comparison. The centerlines (Fig. 4a) and the skewnesses (Fig. 4c) are shown as linear functions of horizontal pixel number. The absolute and relative dispersions (Fig. 4b) are shown on a $\log_{10}$-$\log_{10}$ scale. Overall the behaviour of the centerline, absolute and relative dispersions and skewness calculated from the simulated images is similar to those from the LES densities. The altitude of the centerline is slightly underestimated for horizontal pixels (viewing angles) larger than about 330 ($15°$). Furthermore the skewness from the simulated images differs for horizontal viewing angles $<0°$. The relative and absolute plume dispersion derived from the calculated

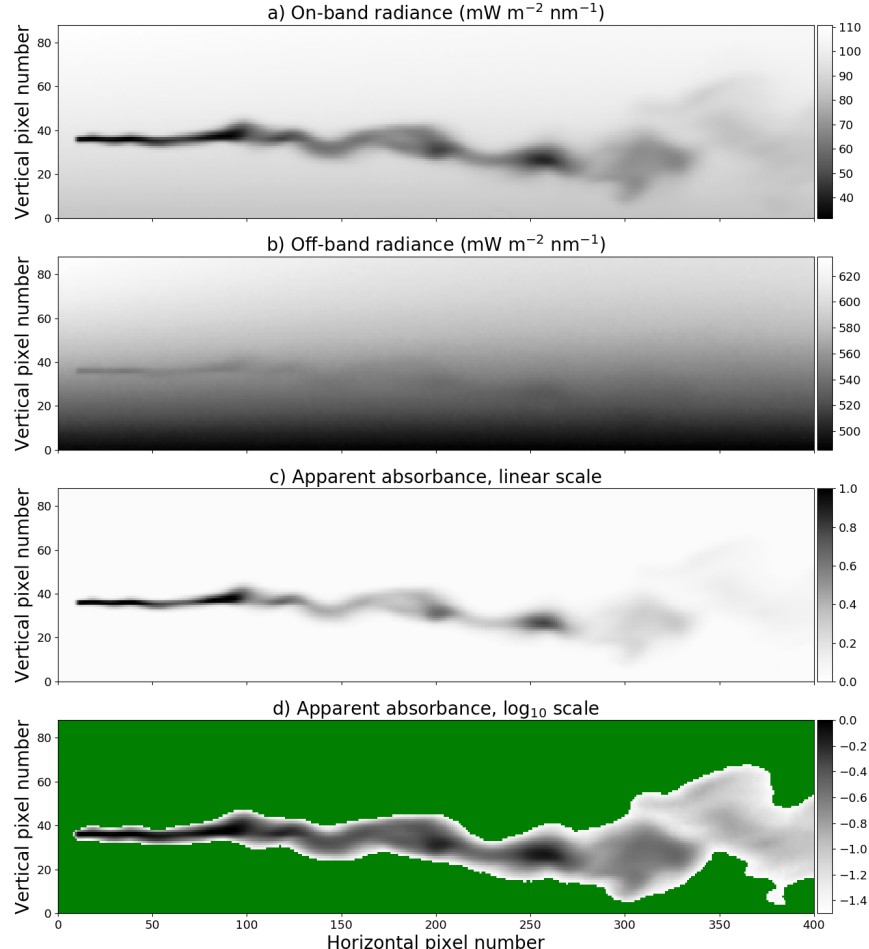

**Figure 3.** Radiances and apparent absorbance for the same time frame as in Fig. 1. (a) The on-band radiance. (b) The off-band radiance. (c) The apparent absorbance calculated using Eq. 2. (d) Same as (c) but with logarithmic grey scale. The green colour represents pixels for which the apparent absorbance $\tau < 0.03$.

images is generally larger than the corresponding LES-values for pixels less than about 100. However, notice that in particular the relative dispersion is smaller than one pixel close to the source, and thus it is highly sensitive to even small differences between the simulated images and the LES concentrations. The MYSTIC Monte Carlo simulations add some noise and thus the dispersion is expected to be larger when derived from the simulated images. For the denser part of the plume the absorption

5   of radiation by $SO_2$ may be saturated (the maximum apparent absorbance is about 1.0), and thus decreasing the skewness. However, inspection of the differences of the third moments without disperion normalization, shows that the differences in skewness is in part driven by the differences in the dispersion. In general, the differences are small and less than one pixel for the entire FOV.



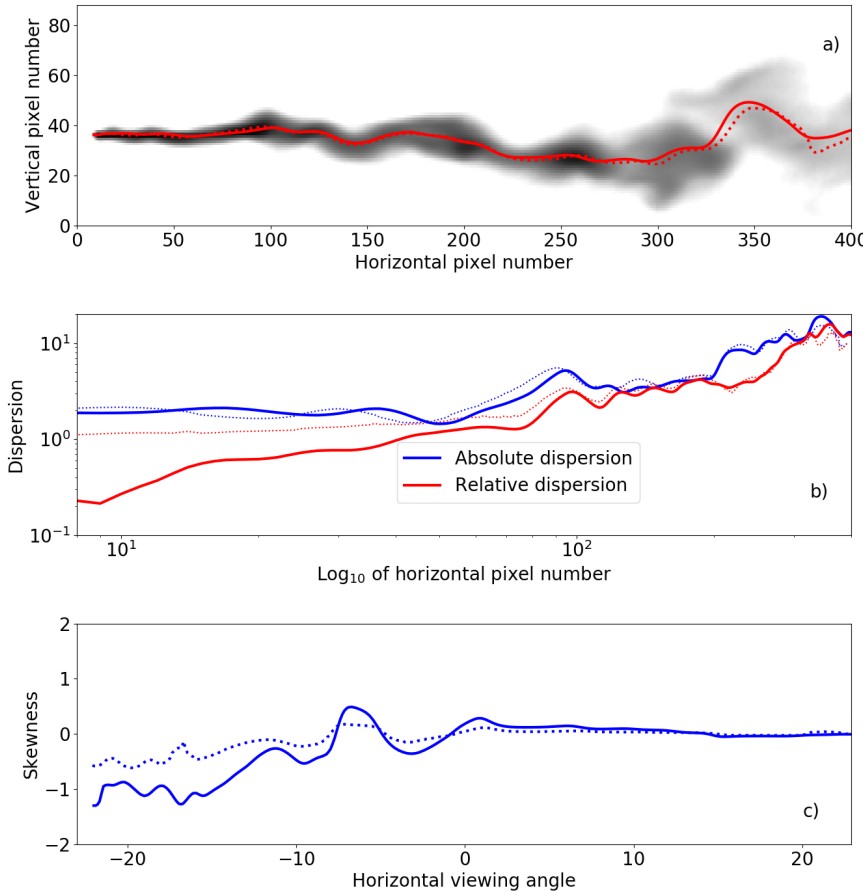

**Figure 4.** (a) The plume apparent absorbance from Fig. 3c with the centerline overlaid in red. (b) The absolute and relative dispersions (note $\log_{10}$ axes). (c) The skewness. Solid lines are calculated directly from the LES densities and shifted to the camera pixels, while dotted lines are derived from the simulated images. Note that the x-axis scales in panels a) and c), while in different units, are the same.

During the COMTESSA campaigns both continuous and puff releases of $SO_2$ are done. For puffs the dispersion increases as the square of the time, $t^2$, of flight (see Dinger et al., 2018, and references therein). Taking the horizontal pixel number as a surrogate for time (Taylor's frozen turbulence hypothesis) we note the linear behaviour of the relative dispersion for pixels close to the release point (Fig. 4b). A linear fit for pixels between 10 and 30 gives slopes of 0.0100 and 0.0217 for the relative dispersion derived from the simulated images and the LES concentrations. If the fit region is changed to pixels between 40 and 60 the slopes changes to 0.0170 and 0.0159 respectively. The results also vary with fit region for other time steps. The relative dispersion is less than one pixel for horizontal pixel numbers smaller than about 60. The differences in slope when varying the





**Table 1.** The mean±the standard deviation for the difference between the simulated images and the LES densities for the time frame shown Fig. 3, and for seven time randomly chosen steps. Numbers are in units of pixels.

| Timestep(s) | Centerline | Absolute Dispersion | Relative Dispersion | Skewness |
|---|---|---|---|---|
| 5 | 0.857±1.372 | -0.152±1.541 | -0.210±1.338 | 0.126±0.282 |
| 5, 13, 22, 31, 41, 60, 97 | 0.521±1.581 | 0.321±1.346 | 0.047±0.870 | 0.053±0.326 |

fit region suggest that for the $t^2$ expansion regime care must be taken when the magnitude of the relative dispersion is small (less than about one pixel) compared to the pixel size of the camera.

As noted above, a large number of images such as in Fig. 3 is required to estimate the parameters of interest for description of turbulence. This is not computationally feasible with available resources. However, to provide an estimate of the difference between the statistical quantities from the simulated images and the LES densities, the differences between the centerline, the absolute and relative dispersions and the skewness, were calculated for seven time steps. The mean differences and the standard deviations are summarized in Table 1. The mean differences between the quantities from the simulated images and the LES densities are on the sub-pixel level. However, the standard deviations of the differences are larger. Possible sources for the differences include Monte Carlo noise in the radiative transfer simulations, image sampling resolution, and errors caused by the spatial resolution of the LES density. It must also be emphasized that the densities simulated by LES and apparent absorbance are not the same physical quantities, but are non-linearly connected through the radiative transfer equation. However, the generally small differences in statistical quantities derived directly from the LES and the radiative transfer model show that apparent absorbance can well serve as a proxy for light-path-integrated density with respect to characterizing turbulent dispersion.

## 3.2 Solar azimuth and zenith angle effects

In Figs. 3 and 4 results were shown for solar azimuth and zenith angles of $\phi_0 = 90°$ and $\theta_0 = 40°$, respectively. The simulations were repeated for a solar zenith angle of $\theta_0 = 60°$ to see if this would change the plume statistics. The difference in apparent absorbance, $\delta\tau = \tau(\theta_0 = 60) - \tau(\theta_0 = 40)$, is shown in Fig. 5. The apparent absorbance is generally slightly smaller (on average about 6%) for $\theta_0 = 60°$ than for $\theta_0 = 40°$. Ideally the apparent absorbance is due to photons travelling along a straight lines passing through the plume and into the camera. However, photons taking other paths may also contribute to the signal. Direct solar radiation photons contribute in three ways to the camera signal through: 1) direct photons scattered behind the plume in the direction of the camera; 2) direct photons scattered in the plume towards the camera; and 3) direct photons scattered between the plume and the camera in the direction of the camera. The first part is included in the apparent absorbance and does not depend on solar zenith angle due to the background correction. The third part is called light dilution and does not depend on the amount of $SO_2$ in the plume. The second part depends on the amount of $SO_2$ in the plume and the solar zenith angle. The latter because there is relatively more direct radiation at $\theta_0 = 40°$ than at $\theta_0 = 60°$. Hence, more direct radiation





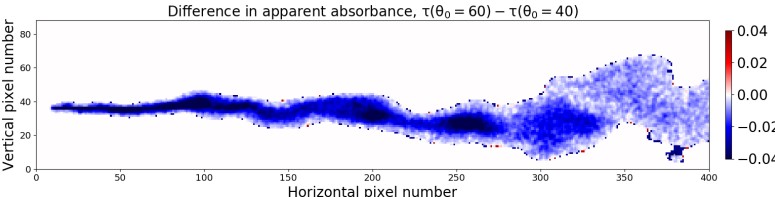

**Figure 5.** The difference in apparent absorbance from images recorded at two different solar zenith angles, $\theta_0 = 60°$ and $\theta_0 = 40°$.

is likely to enter the plume for $\theta_0 = 40°$ and being scattered into the camera from inside the $SO_2$ plume. This explains the negative difference in the apparent absorbance between $\theta_0 = 60°$ and $\theta_0 = 40°$. It is noted that in an experimental setting, where calibrations are carried out throughout the day, solar zenith angle variations will not neccessarily give a change in $SO_2$.

Statistics were calculated as in Fig. 4 for the $\theta_0 = 40°$ case and differences to this case are summarized in Table 2, rows labeled "$\theta_0 = 60°$". In the table, differences are reported as the maximum difference in units of pixel and the percentage of difference values that were larger than one pixel. Differences are reported both without and with the off-band correction, Eqs. 1 and 2 respectively, to quantify the impact of the correction. Overall, the statistics for the $\theta_0 = 60°$ results deviate little from the $\theta_0 = 40°$ case, except for the absolute dispersion, for which 2% of the values differed by more than one pixel. The difference in meandering, not shown in Table 2, is negligible for this and all sensitivity cases below, and is not further discussed.

The sensitivity to the solar azimuth angle was investigated by setting the solar azimuth angle to $\phi_0 = 0, 45, 135, 180$, and $270°$, while keeping the solar zenith angle at $\theta_0 = 40°$. The difference in absorbance is less than 0.05% on average. The impact on the centerline, absolute and relative dispersions and skewness is negligible, see rows labeled "$\phi_0 = 0 - 270°$" in Table 2. From the results no preferable solar azimuth-camera viewing direction geometry may be identified. However, note that the azimuth angle of the background image needs to be the same as for the image with $SO_2$. As no aerosols are included in theses simulations, the differences with and without off-band correction are similar. Finally it is noted that including aerosols has negligible effect on the solar azimuth angle sensitivity, see section 3.3.

## 3.3 Aerosol effects

No aerosols were included in the simulations above. Background aerosols may be present in both the plume and in the surrounding atmosphere. Furthermore, aerosol may be present in the plume due to formation of sulfate aerosol from $SO_2$. Both cases are investigated below.

First, background aerosol with an optical depth $t_{BG}(310) = 0.5$ and a single scattering albedo (SSA) of about 0.95 at 310 nm, were included in simulations for $\theta_0 = 40°$ (The aerosol_default option of uvspec was used, see Emde et al., 2016). The difference between the simulation including background aerosol and the aerosol free simulation is shown in Fig. 6a. Including background aerosol gives generally a slightly lower apparent absorbance, on average about 2.5% whether the off-





**Table 2.** Summary of differences in statistics between the baseline case shown in Fig. 4 and the sensitivity tests. The second column reports the mean and standard deviation (std) of the difference in absorbance ($\delta\tau$). For the centerline, absolute dispersion, relative dispersion and skewness, the mean and standard deviation (std) are given in units of pixels, together with the percentage of pixels where the differences are larger than one pixel (columns labeled %). For the azimuth dependence sensitivity, tests were made for a range of angles, the numbers in the table are the extreme values for all these tests.

| Sensitivity test | $\delta\tau$ | Centerline | | Absolute dispersion | | Relative dispersion | | Skewness | |
|---|---|---|---|---|---|---|---|---|---|
| | Mean±std | Mean±std | % | Mean±std | % | Mean±std | % | Mean±std | % |
| Including off-band correction, Eq.2. | | | | | | | | | |
| $\theta_0 = 60°$ | -0.0097±0.0130 | 0.0151±0.1693 | 0.8 | 0.0545±0.2216 | 1.8 | 0.0580±0.1987 | 1.0 | 0.0017±0.0447 | 0.0 |
| $\phi_0 = 0 - 270°$ | ±0.0001±0.0029 | -0.0048±0.1213 | 0.3 | -0.0079±0.1244 | 0.3 | -0.0098±0.1052 | 0.3 | 0.0022±0.0346 | 0.0 |
| BG aerosol | -0.0044±0.0089 | -0.0143±0.1597 | 0.5 | -0.0111±0.0848 | 0.0 | -0.0199±0.0841 | 0.0 | -0.0007±0.0490 | 0.0 |
| $\tau_{plume} \sim 0.5$, ssa=0.8 | -0.0003±0.0025 | -0.0006±0.0832 | 0.0 | -0.0030±0.0853 | 0.3 | -0.0023±0.0760 | 0.0 | 0.0015±0.0272 | 0.0 |
| $\tau_{plume} \sim 0.5$, ssa=1.0 | -0.0003±0.0024 | -0.0070±0.0918 | 0.3 | -0.0079±0.0988 | 0.5 | -0.0077±0.0825 | 0.3 | 0.0003±0.0236 | 0.0 |
| $\tau_{plume} \sim 5.0$, ssa=0.8 | -0.0035±0.0127 | -0.0280±0.0907 | 0.0 | -0.0072±0.0669 | 0.0 | -0.0073±0.0697 | 0.0 | 0.0018±0.0304 | 0.0 |
| $\tau_{plume} \sim 5.0$, ssa=1.0 | -0.0028±0.0088 | -0.0345±0.0867 | 0.0 | -0.0106±0.0822 | 0.0 | -0.0097±0.0836 | 0.0 | 0.0043±0.0301 | 0.0 |
| A(0.0)-A(0.1) | -0.0013±0.0045 | -0.0015±0.0195 | 0.0 | -0.0110±0.0195 | 0.0 | -0.0132±0.0233 | 0.0 | 0.0005±0.0128 | 0.0 |
| A(1.0)-A(0.1) | 0.0060±0.0172 | -0.0047±0.0334 | 0.0 | 0.0596±0.0516 | 0.0 | 0.0791±0.0610 | 0.0 | 0.0035±0.0223 | 0.0 |
| Not including off-band correction, Eq.1. | | | | | | | | | |
| $\theta_0 = 60°$ | -0.0098±0.0129 | 0.0158±0.1463 | 0.3 | 0.0531±0.1932 | 1.3 | 0.0584±0.1754 | 0.8 | -0.0012±0.0385 | 0.0 |
| $\phi_0 = 0 - 270°$ | -0.0001±0.0027 | -0.0096±0.0752 | 0.0 | -0.0109±0.0818 | 0.0 | -0.0067±0.0791 | 0.3 | 0.0011±0.0312 | 0.0 |
| BG aerosol | -0.0045±0.0089 | -0.0109±0.1016 | 0.0 | -0.0114±0.0804 | 0.0 | -0.0053±0.0757 | 0.0 | -0.0007±0.0272 | 0.0 |
| $\tau_{plume} \sim 0.5$, ssa=0.8 | 0.0021±0.0033 | -0.0038±0.0361 | 0.0 | -0.0088±0.0499 | 0.0 | -0.0107±0.0503 | 0.0 | 0.0002±0.0232 | 0.0 |
| $\tau_{plume} \sim 0.5$, ssa=1.0 | 0.0012±0.0024 | -0.0072±0.0706 | 0.0 | -0.0072±0.0706 | 0.0 | -0.0119±0.0603 | 0.0 | -0.0010±0.0229 | 0.0 |
| $\tau_{plume} \sim 5.0$, ssa=0.8 | 0.0199±0.0263 | -0.0533±0.1480 | 0.3 | -0.0696±0.1604 | 0.03 | -0.0928±0.1827 | 1.3 | -0.0004±0.0521 | 0.0 |
| $\tau_{plume} \sim 5.0$, ssa=1.0 | 0.00109±0.0139 | -0.0629±0.1343 | 0.5 | -0.0479±0.1140 | 0.3 | -0.0595±0.1224 | 0.3 | 0.0020±0.0415 | 0.0 |
| A(0.0)-A(0.1) | -0.0013±0.0043 | 0.0003±0.0158 | 0.0 | -0.0103±0.0181 | 0.0 | -0.0131±0.0198 | 0.0 | 0.0005±0.0082 | 0.0 |
| A(1.0)-A(0.1) | 0.0057±0.0171 | -0.0033±0.0330 | 0.0 | 0.0626±0.0521 | 0.0 | 0.0822±0.0615 | 0.0 | 0.0034±0.0223 | 0.0 |



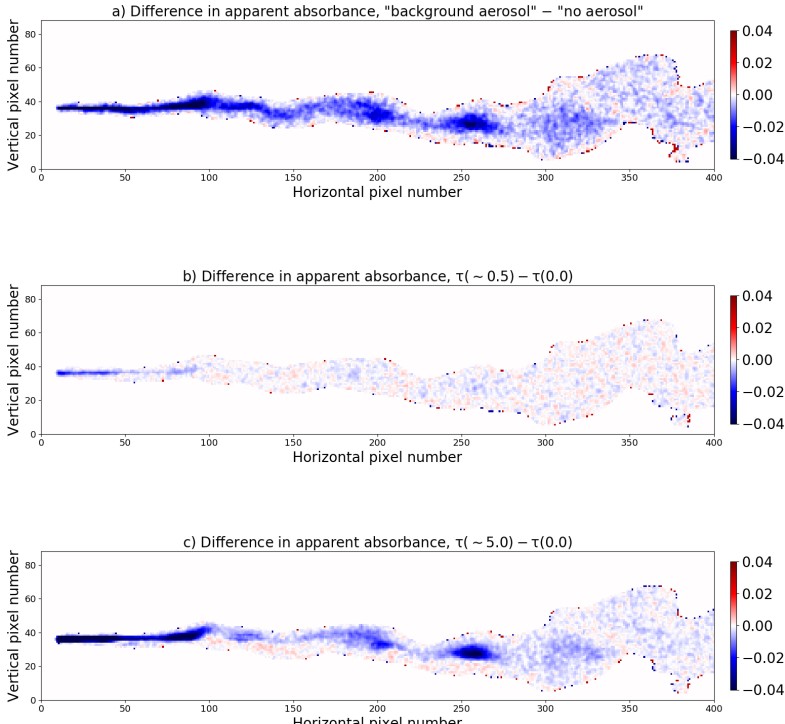

**Figure 6.** The difference in apparent absorbance from images recorded (a) with and without background aerosols in the atmosphere; (b-c) with and without aerosol in the plume (b): $\tau_{plume} \sim 0.5$; (c): $\tau_{plume} \sim 5.0$.

band correction is excluded or included, Eqs. 1 and 2 respectively. The decrease is due to multiple scattering by the aerosol and hence less direct radiation (Kern et al., 2010). The background aerosol had negligible effect on plume statistics, Table 2, rows labeled "BG aerosol".

Simulations were also made with aerosol in the plume only, that is various amounts of aerosol were added to the voxels containing $SO_2$. This is relevant for non-pure $SO_2$ plumes where aerosols are co-emitted such as from power plants, or where secondary sulfate aerosols may form in the $SO_2$ plume. The impact of both highly absorbing (SSA=0.8) and purely scattering aerosol (SSA=1.0) were investigated. Kern et al. (2013) concluded that if a plume contains an absorbing aerosol component the retrieved $SO_2$ columns may be underestimated. Here we estimate the effect of aerosols in the plume on the higher order statistics of the plume. Figs. 6b and 6c show the difference in apparent absorbance between simulations with and without absorbing aerosol in the plume for a relatively large aerosol optical depth of about 0.5 (Fig. 6b) and an unrealistic extreme case with aerosol optical depth of about 5.0 at 310 nm (Fig. 6c). Results for non-absorbing aerosol are similar (not shown).

For the more realistic value, $\tau_{plume} \sim 0.5$, Fig. 6b, there is little impact on the apparent absorbance. For (non-)absorbing aerosol with ssa=0.8 (ssa=1.0) the decrease is less than 0.2% (0.16%) on average with off-band correction, and the increase is





less than 1.1% (0.63%) on average without off-band correction. For (unrealistically) large amounts of (non-)absorbing aerosol in the plume, $\tau_{plume} \sim 5.0$, the apparent absorbance decreases by less than 2% (1.5%) on average if off-band correction is included, Eq. 2. Without off-band correction, Eq. 1, the apparent absorbance increases by 9.5% (5.43%) on average. For all cases, the off-band correction reduces the influence of aerosol, as intended. For both aerosol in plume cases and whether the

off-band correction was included or not, the plume statistics were affected to a negligible extent, as reported in rows labeled "$\tau_{plume}$" in Table 2.

The sensitivity of the solar azimuth angle when including aerosols was investigated by performing additional simulations for $\phi_0 = 45$ and $180°$ for the aerosol in the plume and background aerosol cases with $\tau_{plume} \sim 0.5$. The solar azimuth angle sensitivity for these cases were of the same magnitude as for the aerosol free simulations and thus of negligible impact.

**3.4   Surface albedo**

All simulations above were made with a surface albedo $A = 0.0$ to avoid coupling between the various processes that affect the camera images. For the wavelengths considered here the albedo for snow-free surfaces is generally small ($A < 0.1$, see for example Wendisch et al., 2004). To test the sensitivity to snow free surface albedo, simulations were made for surface albedos of $A = 0.05$ and $A = 0.1$. In addition a simulation was made with $A = 1.0$ to estimate the effect of fresh snow which

has a an albedo close to one at UV wavelengths (Wiscombe and Warren, 1980). The background images were calculated for each individual case. The apparent absorbance difference for the A(0.0)-A(0.1) and A(1.0)-A(0.1) cases are shown in Fig. 7. The overall results are summarized in Table 2. Decreasing the albedo from 0.1 to 0.0 gives an overall reduction in the

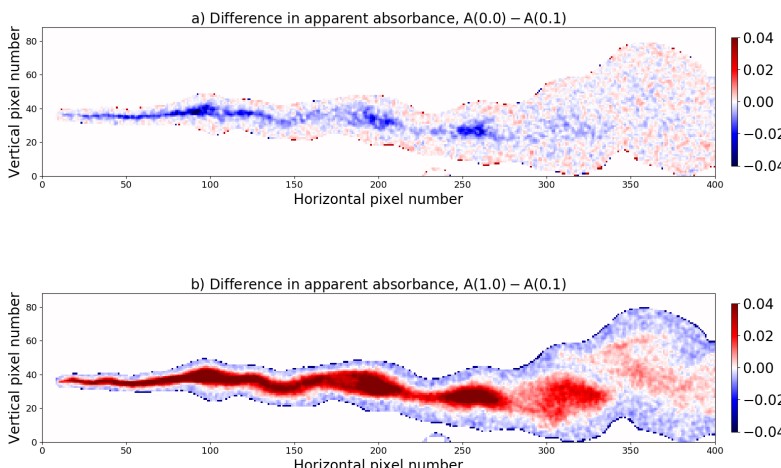

**Figure 7.** The difference in apparent absorbance from images recorded with surface albedo of (a) A(0.0) and A(0.1) and (b) A(1.0) and A(0.1).





apparent absorbance (mostly blue colors in Fig. 7a). Compared to the $A = 0.1$ case , the $A = 0.05$ case, not shown, is about a factor of 2 smaller in magnitude for the mean apparent absorbance. Increasing the albedo from 0.1 to 1.0 gives an increase in the apparent absorbance (mostly red colors in Fig. 7b). As mentioned above, section 3.2, the apparent absorbance is due to photons travelling along straight lines passing through the plume and into the camera. However, photons scattered between

5 the plume and the camera, and multiple scattered photons within the plume, termed the light dilution and multiple scattering effect, respectively, may distort the apparent absorbance (Kern et al., 2012). An additional distortion, not discussed by Kern et al. (2012), is due to the surface albedo which gives additional photon paths that may contribute to the camera signal. Some photons may scatter off the surface into the plume and in the direction of the camera. This will give increased (decreased) apparent absorbance with increasing (decreasing) albedo for relatively large $SO_2$ concentration, see red (blue) signal in Fig. 7b

10 (a). For small $SO_2$ concentrations, the light dilution effect prevails, giving a reduction (increase) in the apparent absorbance for increasing (decreasing) albedo, see blue (red) signal in Fig. 7b (a). While albedo changes may both increase and decrease the apparent absorbance, the impact on plume statistics is minor. Thus, overall, the surface albedo has negligible effect on the plume statistics (Table 2).

### 3.5 Fractal dimension

15 The mass box counting method, see section 2.3.2, was applied to seven random time frames (see Table 1) of LES-densities as exemplified in Fig. 1b and corresponding apparent absorbances from the radiative transfer simulation images for the reference situation (Fig. 3c). The mean $D_{box}$ values for all LES-densities and radiative transfer simulation images were 1.170 and 1.156, respectively. In general there is no simple relationship between trace gas concentrations and the camera radiances as these are connected via the radiative transfer equation. Hence, the fractal dimensions of concentration fields and images may differ. The

20 slightly lower dimension of the images may indicate that some radiative smoothing effect implies a loss of heterogeneity.

  We note that Sykes and Gabruk (1994) report fractal dimension between 1.3 and 1.35 from area-perimeter and box-counting analysis for LES calculations representing neutral and convective conditions. Their results are for a different cross-section of the plume than the results presented here, namely the y-z plane in our coordinate system, Fig. 2. Furthermore, we analyse integrated line paths while they studied concentration. Thus, in their case a more heterogeneous surface may be expected.

25 ## 4 Conclusions

Turbulence is one of the unsolved problems of physics. One novel method to measure atmospheric turbulent tracer dispersion is to use UV cameras sensitive to absorption of sun-light by $SO_2$. In this paper we have presented a method to simulate such UV camera measurement with a 3D Monte Carlo radiative transfer model. Input to the radiative transfer simulations are large eddy simulations (LES) of a $SO_2$ plume. From the simulated images various plume density statistics (centerline position, me-

30 andering, absolute and relative dispersions, skewness, fractal dimension) were calculated and compared with similar quantities directly from the LES. Mean differences between the simulated images and the LES were found to be smaller than a quarter of a pixel, with standard deviations between 1/2 and 3/2 pixel.



Furthermore sensitivity studies were made to quantify how changes in solar azimuth and zenith angles, aerosol (background and in plume), and surface albedo, impact the UV camera image plume statistics. It was found that changing the parameters describing these effects within realistic limits, had negligible effect on the centerline position, meandering, absolute and relative dispersions, and skewness of the $SO_2$ plume.

5 Thus, based on the simulated UV camera images and the comparison with the LES, it can be concluded that UV camera images of $SO_2$ plumes may be used to derive plume statistics of relevance for the study of atmospheric turbulence.

*Code availability.* The libRadtran software used for the radiative transfer simulations is available from www.libradtran.org. The PALM model system was used for the LES and it is available from palm.muk.uni-hannover.de/trac. The ImageJ software is available from https://imagej.nih.gov/ij/ and the FracLac plugin from https://imagej.nih.gov/ij/plugins/fraclac/FLHelp/Introduction.htm.

10 *Author contributions.* AK performed the radiative transfer simulations. HA, MC and SYP were responsible for the LES. AK prepared the manuscript with contributions from all co-authors.

*Competing interests.* The authors declare that no competing interests are present.

*Acknowledgements.* The Comtessa project has received funding from the European Research Council (ERC) under the European Union's Horizon 2020 research and innovation programme under grant agreement no. 670462.



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
