# Peer review of "Can statistics of turbulent tracer dispersion be inferred from camera observations of $SO_2$ in the ultraviolet? A modelling study"

_Atmospheric Measurement Techniques, 2019_

## Referee Comment (RC1) · Anonymous Referee #1 · 6 Sep 2019

This paper models an elevated, continuous release of SO2 gas in a channel flow using large-eddy simulation (LES). It uses a radiative transfer model to model what an ultraviolet camera would see and compare that with line-of-sight integrated SO2 directly from the LES. Several statistics are compared such a plume dispersion, both relative and absolute, and it is shown that the statistics of those does not change. There are no comparison with real measurements data of any kind.

Generally speaking, the LES with the setup that the authors have chosen is not adequate to simulate plume dispersion. The grid resolution is around 1 m in all three directions and it is well known that only eddies of the size of almost ten grid cells are

well resolved. The authors use the plume from it is released until 200 meters downstream for their investigation. Even 125 m downstream from the source the plume is not larger than ten times the grid resolution which means that the plume is not dispersing due to turbulence but rather because of the sub-grid-scale parametrisation of the LES, which I assume it more like molecular diffusion (The authors do not describe that process in detail). This means that most of the plume does not look like real dispersion but appears much smoother. It is therefore dubious to replace experimental measurements with LES in this case. They simulate and store 100 snapshots of the plume evolution.

Then the authors go to radiative transfer calculations with the sun as the light source. It is rather surprising that the calculations use circular boundary conditions such that when one photon leaves the domain on one side it appears again on the opposite side. The refer to energy conservation for this, but I simply don't understand the reasoning behind. It appears unphysical and gives rise to various artefacts such as ghost plumes that have to be removed. They analyse only one (!) out of the 100 snapshots which I think is a very little number for getting good statistics.

The plume statistics analysis is a bit messy. There is an equivalence between the x and z position in space and the two pixel coordinates in the camera. This might be a good approximation but it is confusing that the same symbols are used for the different physical quantities. The mean plume height is a mean of plume heights at all downstream positions. Usually, in a boundary layer the mean plume height is a function of downstream position x since the plume may rise if it is released close to the surface. As a consequence of choosing the average plume height over all x values is that the absolute dispersion sigma_z is a mix of ordinary absolut dispersion (which is relative to the mean height at a specific downstream position x) and the general plume rise. The definition of meandering dispersion suffers some of the same inconveniences. The poor reason for those somewhat awkward definitions is that the authors have an ensemble of one which prevents making ensemble means as is usually done.

Jumping to fractal dimensions, the authors fail to describe exactly what the do, and one could ask how relevant fractal dimensions are given the poor resolution of the LES. Often fractal dimensions are used to describe the interface between the plume and the surrounding air, but here it is unclear what N(epsilon) really is. There is no supporting figure to let the reader know.

The lack of realism of the LES of the plume is also displayed in figure 4 where the relative dispersion is shown. The slope of the red curves in this double logarithmic plot around 1/2 indicating pure molecular-like dispersion (if it is sigma_zr which is plotted) . It is confusing that the authors talk about slope between 0.01 and 0.0217 while I get something from the plot around 1/2. Theoretically, the slope in this range should be close to 3/2 as also mentioned in the Dinger et al paper which they refer to.

To summarize, it looks like the work does not spend enough computational resources on doing a realistic dispersion simulation and, secondly, doing analysis of all their snapshots to get proper statistics.

"Turbulence is one of the unresolved problems of physics" it is written at several occasions. It is not very clear that this work brings us much further.

---

## Referee Comment (RC2) · Anonymous Referee #2 · 11 Nov 2019

Brief summary of the paper...

The aim of the paper is to present a novel method to simulate UV camera images of a dispersing SO2 plume with a 3D Monte Carlo radiative transfer model, and then to examine how various factors (solar angles, aerosol content, and surface albedo) affect the statistical parameters characterizing plume dispersion. Instead of a real atmospheric flow, results from a large eddy simulation (LES) of a plume are used to validate the simulation method. The success of the simulated UV camera images is assessed by calculating plume statistics (first three moments of the concentration field in the absolute and relative coordinate systems) and comparing results for the LES data with

those for the simulated images. Tracer dispersion was further analyzed by calculating the fractal dimension using a mass box counting method.

Several reference cases were considered and analysed in detail, yielding good results. Based on these results, the authors conclude that UV camera images of SO2 plumes in real atmospheric turbulence may be used to make quantitative investigations of plume dispersion in the real atmosphere.

1. General comments (overall quality)

The approach taken by the authors strikes me as innovative and valuable. It is clear to me that the good results for the plume statistics comparisons gives needed confidence that satellite UV camera images can potentially provide accurate fluid dynamical information.

Having said that, I have some criticisms of the data analysis.

2. Specific comments (scientific questions / issues)

One criticism is that the usefulness of the fractal dimension calculation is unclear. Indeed, the authors description of the mass box counting method lacks detail, giving the impression that they do not know how best to make use of this parameter.

Another is that a more complete comparison could be made by comparing concentration PDFs. PDFs may or may not yield interesting information for the LES and simulated images used in this study, but in real turbulence one often finds intermittency, and its signature can be seen in the tails of the PDFs.

Finally, an analysis of the LES velocity field, projected onto the planes of the camera images (2D slices of the field), was not done. Calculations of divergence, vorticity and rate of strain in these 2D slices will help to identify vortex cores, saddle points and, where large 2D divergence will show up regions where out-of plane motion is significant. Such information will help interpret the structure of the concentration field and tracer dispersion, and the experience should generate intuition useful to the interpretation of images from real atmospheric flows.

3. Technical corrections.

In the Introduction, please explain to the reader the motivation to investigate SO2 instead of some other tracer(s).

Is this the first time a simulation of camera images or UV camera images have been attempted? If so, please say. If there have been previous efforts, were they successful or not?

Page 2, line 17: The phrase "based on a large eddy simulation (LES)" does not convey the correct impression. I think you mean to say that you use LES in lieu of a real atmospheric flow. The following sentence appears twice, once in the abstract and once in the conclusions. "Turbulence is one of the unsolved problems of physics." In both cases the sentence is unnecessary and distracting. It should be deleted.

A similar sentence appears in the Introduction (line 8): "The complete description of turbulence remains one of the unsolved problems of physics." This sentence also seems out of place and unnecessary and should be deleted.
* * *

---

## Author Comment (AC1) · 31 Mar 2020

**Response to interactive comments from Referee #1**

We are a little surprised by the comments by the referee and realise that we have somehow failed to communicate what the manuscript aims to present. To this end we have revised the title, abstract, introduction and conclusions to clarify the manuscript. Besides adressing the comments, we have also made the following changes to the manuscript:

- In the previous version of the manuscript all results were reported in the units of camera pixels. We now report in physical units (meters) where appropriate.

- To obtain statistics for locations further downwind from the release point, we have included results for three more camera positions. One, camera B, at the same location as the original camera A, but with a smaller horizontal field of view, and similar cameras at about 300 m (camera B), and 500 m (camera C) downwind from the release point. For cameras B, C and D the release point of the plume is at a higher altitude to allow the cameras to see the full vertical extent of the plume. Furthermore, these cameras have more pixels in the vertical direction than camera A.

- To be able to compare the LES with the simulated images for these new camera viewing directions (viewing the plume at an elevation angle of 30.7° compared to 5.7°) new software had to be developed to calculate column densities along the camera line of sight through the LES 3D concentration.

The referee give several comments in a non-listed format. Below we have extracted the comments from the referee response and answered them one by one. The referee's comments are in italic font. The responses to the comments are shown in roman font.

**Comments**

- *There are no comparison with real measurements data of any kind.*

  This is modelling study with the aim to investigate whether it is feasible to derive statistics from UV camera images or not. To make this clear, we have changed the title to: 'Can statistics of turbulent tracer dispersion be inferred from camera observations of $SO_2$ in the ultraviolet? A modelling study'.

- *Generally speaking, the LES with the setup that the authors have chosen is not adequate to simulate plume dispersion. The grid resolution is around 1 m in all three directions and it is well known that only eddies of the size of almost ten grid cells are well resolved.*

  We investigated effects of grid resolution on plume dispersion in great detail in a recent work (Ardeshiri et al., 2020) where we used also higher resolutions. Indeed, the somewhat inadequate resolution near the source generates a larger effective source size by numerical diffusion. Unfortunately, these high-resolution simulations cannot be used because of the memory requirements of the radiative transfer model. Radiative transfer is non-local in nature and the full domain must be in memory for calculations to be efficient. Increasing the spatial resolution by a factor of 10, in 3D increase the memory demand by a factor of 1000, which is not available to us. However, the purpose of this paper is not to provide the most realistic LES results, but to demonstrate that integrated characteristic of the plume can be reconstructed accurately from camera data. And for showing this, the LES that we use is sufficient.

- *The authors use the plume from it is released until 200 meters down-stream for their investigation. Even 125 m downstream from the source the plume is not larger than ten times the grid resolution which means that the plume is not dispersing due to turbulence but rather because of the sub-grid-scale parametrisation of the LES, which I assume it more like molecular diffusion (The authors do not describe that process in detail). This means that most of the plume does not look like real*

Table 1: The mean±the standard deviation for the difference between the simulated images and the LES densities for seven (camera A) and nine (cameras B, C and D) randomly chosen time steps.

| Camera | Timestep(s) | Centerline (m) | Absolute dispersion (m) | Relative dispersion (m) | Skewness |
|---|---|---|---|---|---|
| A | 5, 13, 22, 31, 41, 60, 97 | 0.060±0.203 | 0.955±0.655 | 1.243±0.602 | -0.029±0.094 |
| B | 7, 23, 32, 48, 60, 61, 65, 75 91 | -0.503±1.164 | -1.088±1.612 | -0.240±0.565 | -0.187±0.252 |
| C | 7, 23, 32, 48, 60, 61, 65, 75 91 | -1.069±2.582 | 4.989±2.141 | 4.429±2.030 | -0.159±0.794 |
| D | 7, 23, 32, 48, 60, 61, 65, 75 91 | -0.353±2.851 | 7.658±1.839 | 7.036±1.681 | 0.505±1.375 |

*dispersion but appears much smoother. It is therefore dubious to replace experimental measurements with LES in this case.*

We are well aware of the effects of numerical and sub-grid-scale diffusivity, see also answer to previous question. Furthermore we have revised our results to use world coordinates instead of pixel coordinates. Revised results for the camera in the original manuscript are shown in Fig. 1. We note that at the horizontal angle of 90°, corresponding to about 110 m downstream from the release point, the plume is already about $4\text{-}5\times\sigma_r \approx 16-20$ m. Thus, already at 110 m downstream the actual plume size is much larger than the LES voxel size.

[Figure]

Figure 1: a) The plume apparent absorbance. b) The LES column density integrated along the line of sight. Colorscales in a) and b) are relative and thus no colorbars are provided. c) The centerline, absolute and relative dispersions. d) The skewness. The solid lines are LES results and dotted lines are values calculated from the simulated images.

We have made simulations for cameras seeing the plume further downstream, Fig. 2.

One example for camera D is shown in Fig. 3. We have updated the table describing the reference comparison to include the results for the new cameras, see Table 1. The manuscript have been updated with these new results and corresponding discussions.

Finally, the sub-grid-scale treatment of the LES is described by the following senence in section 2.1 of the manuscript:

[Figure]

Figure 2: Bird's eyes view of the 3D domain (black square) and the $SO_2$ plume location within the domain (red square, for camera A simulation, shifted along x-axis for the other cameras). The UV cameras are located where the two green or blue lines intersect. The lines indicate the horizontal field-of-view of the cameras. The column density of the plume is included for illustrative purpose. The direction of the incoming Sun ray is shown by the yellow line.

> In this methodology, the large scales of the turbulent flow are explicitly simulated while a low-pass filter is applied to the governing equations to remove the small scales information from the numerical solution. The effects of the small scales are then parameterized by means of a sub-grid scale (SGS) model (e.g. Deardorff, 1973; Moeng, 1984; Pope, 2000; Celik et al., 2009).

- *Then the authors go to radiative transfer calculations with the sun as the light source. It is rather surprising that the calculations use circular boundary conditions such that when one photon leaves the domain on one side it appears again on the opposite side. The refer to energy conservation for this, but I simply dont understand the reasoning behind. It appears unphysical and gives rise to various artefacts such as ghost plumes that have to be removed.*

The radiative transfer model used is based on the Monte Carlo technique and trace photons through the model-domain (the atmosphere). The model-domain is a rectangular cuboid. With a solar source the photons enter the domain through the top surface. Photons are traced until they leave the domain or are absorbed. If a photon leaves the domain through one of the four sides it is lost and will remain unaccounted for, and hence energy conservation is broken which might lead to non-physical results. Thus, to quote Mayer (2009):

> If a photon leaves the model domain through the side, we apply periodic boundary conditions: the photon re-enters at the opposite side; this ensures energy conservation and is appropriate for most applications, given that the model domain is large enough so that the process under consideration are not affected by edge effects.

- *They analyse only one (!) out of the 100 snapshots which I think is a very little number for getting good statistics.*

This statement is not correct. In Table 1 of the manuscript statistics based on 7 snapshots are presented. In addition we have now also performed simulations for more cameras, see Table 1. For the sensitivity analysis, however, only one snapshot is used. Though, there is no physical reason

[Figure]

Figure 3: Results for Camera D and time step 91. a) The plume apparent absorbance. b) The LES column density integrated along the line of sight. Colorscales in a) and b) are relative and thus no colorbars are provided. c) The centerline, absolute and relative dispersions. d) The skewness.

for the results from the sensitivity analysis to be different if other or more snapshots are used. Also, note that if the instantaneous spatial statistics are correct, the ensemble statistics will also be correct. This is not necessarily true the other way around. We have revised the abstract and the introduction part of the manuscript to make this clearer.

- *The plume statistics analysis is a bit messy. There is an equivalence between the x and z position in space and the two pixel coordinates in the camera. This might be a good approximation but it is confusing that the same symbols are used for the different physical quantities. The mean plume height is a mean of plume heights at all downstream positions. Usually, in a boundary layer the mean plume height is a function of downstream position x since the plume may rise if it is released close to the surface. As a consequence of choosing the average plume height over all x values is that the absolute dispersion sigma$_z$ is a mix of ordinary absolut dispersion (which is relative to the mean height at a specific downstream position x) and the general plume rise. The definition of meandering dispersion suffers some of the same inconveniences. The poor reason for those somewhat awkward definitions is that the authors have an ensemble of one which prevents making ensemble means as is usually done.*

In the revised manuscript we do not use pixel coordinates, but world coordinates $x$ and $z$ throughout. Furthermore, as pointed out by the referee: The absolute dispersion can be properly defined only using an ensemble or time average. This is not possible here as we do not have an esemble available. We thus adopt the center of the source ($\bar{z} = z_0$) as the reference vertical position and define the absolute dispersion accordingly. It is noted, that with this definition of the absolute position, relative and absolute dispersion are correctly the same at the source location, since meandering

here is zero. The revised statistics are shown in Fig. 1.

- *Jumping to fractal dimensions, the authors fail to describe exactly what they do, and one could ask how relevant fractal dimensions are given the poor resolution of the LES. Often fractal dimensions are used to describe the interface between the plume and the surrounding air, but here it is unclear what N(epsilon) really is. There is no supporting figure to let the reader know.*

  In response to the comment we have decided to leave this section out.

- *The lack of realism of the LES of the plume is also displayed in figure 4 where the relative dispersion is shown. The slope of the red curves in this double logarithmic plot around 1/2 indicating pure molecular-like dispersion (if it is sigma$_{zr}$ which is plotted). It is confusing that the authors talk about slope between 0.01 and 0.0217 while I get something from the plot around 1/2. Theoretically, the slope in this range should be close to 3/2 as also mentioned in the Dinger et al paper which they refer to.*

  In response to the comment we have decided to leave this section out.

- *To summarize, it looks like the work does not spend enough computational resources on doing a realistic dispersion simulation and, secondly, doing analysis of all their snap-shots to get proper statistics.*

  As discussed briefly above, and fully discussed in Ardeshiri et al. (2020), the LES simulations are not perfect, despite this the grid resolution preserve qualitative characteristic similar to more resolved simulations. Indeed in Ardeshiri et al. (2020) it is demonstrated that concentration PDF of the coarser resolution simulations has the same shape but with less fluctuations. As mentioned in the manuscript, and further explained above, it is not computationally feasible to perform radiative transfer simulations for all snap-shots. Nor is it required to reach the aims of this manuscript. Both these points have been emphasized in the revised manuscript.

- *Turbulence is one of the unresolved problems of physics it is written at several occasions. It is not very clear that this work brings us much further.*

  This sentence has been removed throughout the manuscript.

**Bibliography**

Ardeshiri, H., Cassiani, M., Park, S., Stohl, A., I.Pisso, and Dinger, A.: On the convergence and capability of large eddy simulation for passive plumes concentration fluctuations in an infinite-Re neutral boundary layer, Boundary-Layer Meteorol., accepted for publication, 2020.

Celik, I., Klein, M., and Janicka, J.: Assessment measures for engineering les application, J. Fluid Eng, 131(3), 031 102, 2009.

Deardorff, J.: The use of subgrid transport equations in a three-dimensional model of atmospheric turbulence, J. Fluid Eng, 95, 429–438, 1973.

Mayer, B.: Radiative transfer in the cloudy atmosphere, Eur. Phys. J. Conferences, 1, 75–99, 2009.

Moeng, C.: A large-eddy simulation model for the study of planetary boundary-layer turbulence, J. Atmos. Sci, 41, 2052–2062, 1984.

Pope, S. B.: Turbulent Flows, Cambridge University Press, 2000.

---

## Author Comment (AC2) · 31 Mar 2020

**Response to interactive comments from Referee #2**

Besides adressing the comments, we have also made the following changes to the manuscript:

- In the previous version of the manuscript all results were reported in the units of camera pixels. We now report in physical units (meters) where appropriate.

- To obtain statistics for locations further downwind from the release point, we have included results for three more camera positions. One, camera B, at the same location as the original camera A, but with a smaller horizontal field of view, and similar cameras at about 300 m (camera B), and 500 m (camera C) downwind from the release point. For cameras B, C and D the release point of the plume is at a higher altitude to allow the cameras to see the full vertical extent of the plume. Furthermore, these cameras have more pixels in the vertical direction than camera A.

- To be able to compare the LES with the simulated images for these new camera viewing directions (viewing the plume at an elevation angle of 30.7° compared to 5.7°) new software had to be developed to calculate slant column densities along the camera lines of sight through the LES 3D concentration.

Below the comments from Referee #2 are given in italic font. Our responses to the comments are shown in roman font.

**Specific comments**

- *One criticism is that the usefulness of the fractal dimension calculation is unclear. Indeed, the authors description of the mass box counting method lacks detail, giving the impression that they do not know how best to make use of this parameter.*

  In response to the comment we have decided to leave this section out.

- *Another is that a more complete comparison could be made by comparing concentration PDFs. PDFs may or may not yield interesting information for the LES and simulated images used in this study, but in real turbulence one often finds intermittency, and its signature can be seen in the tails of the PDFs.*

  The images provides column densities (typically units of $1/m^2$) along the line of sight of the camera. This is a quantity different from the concentration (units of $1/m^3$) usually used to calculate PDFs. We have, however, calculated the probability density function (pdf) of the column densities from the LES and simulated images as shown in Fig. A1. These pdfs and their contribution to the explanation of the differences between the statitistical quantities from the LES and simulated images, are discussed in the revised manuscript.

[Figure]

Figure A1: The probability density function (pdf) of the column densities from the LES and simulated images in Figs. 4-7 in the revised manuscript.

- *Finally, an analysis of the LES velocity field, projected onto the planes of the camera images (2D slices of the field), was not done. Calculations of divergence, vorticity and rate of strain in these 2D slices will help to identify vortex cores, saddle points and, where large 2D divergence will show up regions where out-of plane motion is significant. Such information will help interpret the structure of the concentration field and tracer dispersion, and the experience should generate intuition useful to the interpretation of images from real atmospheric flows.*

  A velocity field can be extracted from the camera images using for example the techniques described by Gliß et al. (2017). However, this velocity field is integrated by the concentration along the view path. As such it is very different from 2D slices and a comparison between the two is not trivial nor straightforward. The LES velocity and concentration fields are thoroughly discussed in the paper by Ardeshiri et al. (2020) which is cited in the manuscript.

**Technical corrections**

- *In the Introduction, please explain to the reader the motivation to investigate SO2 instead of some other tracer(s).*

To motivate the use of SO$_2$ we have added the following text to the Introduction:

> Over short transport distances, sulfur dioxide (SO$_2$) may be considered to be a passive tracer. Furthermore, SO$_2$ strongly absorbs radiation in part of the UV spectrum and may thus be detected by for example UV sensitive cameras (see for example Kern et al., 2010, and references therein).

- *Is this the first time a simulation of camera images or UV camera images have been attempted? If so, please say. If there have been previous efforts, were they successful or not?*

  To the authors knowledge UV camera images have not been simulated before. This statement has been added to the Introduction.

- *Page 2, line 17: The phrase "based on a large eddy simulation (LES)" does not convey the correct impression. I think you mean to say that you use LES in lieu of a real atmospheric flow.*

  We have rephrased this part so it now reads:

  > We present a novel method to simulate UV camera images of a dispersing SO$_2$ plume using a 3D radiative transfer model. The 3D descripton of the SO$_2$ plume is provided by large eddy simulation (LES) and are used in lieu of real atmospheric flow. The simulated images are used to examine how various factors (solar angles, aerosol content, and surface albedo) affect the statistical parameters characterizing the SO$_2$ plume dispersion.

- *The following sentence appears twice, once in the abstract and once in the conclusions. "Turbulence is one of the unsolved problems of physics." In both cases the sentence is unnecessary and distracting. It should be deleted.*

  This sentence has been removed both places as suggested.

- *A similar sentence appears in the Introduction (line 8): "The complete description of turbulence remains one of the unsolved problems of physics." This sentence also seems out of place and unnecessary and should be deleted.*

  The sentence has been removed as suggested.

**Bibliography**

Ardeshiri, H., Cassiani, M., Park, S., Stohl, A., I.Pisso, and Dinger, A.: On the convergence and capability of large eddy simulation for passive plumes concentration fluctuations in an infinite-Re neutral boundary layer, Boundary-Layer Meteorol., accepted for publication, 2020.

Gliß, J., Stebel, K., Kylling, A., Dinger, A. S., Sihler, H., and Aasmund, S.: Pyplis - A Python Software Toolbox for the Analysis of SO2 Camera Images for Emission Rate Retrievals from Point Sources, Geosciences, 7, https://doi.org/10.3390/geosciences7040134, URL http://www.mdpi.com/2076-3263/7/4/134, 2017.

Kern, C., Kick, F., Lübcke, P., Vogel, L., Wöhrbach, M., and Platt, U.: Theoretical description of functionality, applications, and limitations of $SO_2$ cameras for the remote sensing of volcanic plumes, Atmospheric Measurement Techniques, 3, 733–749, https://doi.org/10.5194/amt-3-733-2010, URL http://www.atmos-meas-tech.net/3/733/2010/, 2010.

---

## Author Response (AR2)

**Response to report from the Associate Editor**

In addition to the changes in response to the comments from the referees, we have made the following changes:

- Updated the affiliations of Andreas Stohl, Hamidreza Ardeshiri, and Anna Solvejg Dinger.

- Added an acknowledgement.

Below the referee's comments are in italic font. The responses to the comments are shown in roman font.

**Response to comments from Referee #1**

**Comments**

*I am still not completely convinced about the circular boundary conditions in the radiative transfer calculations. Does it effectively mean that their are infinitely many plumes in the horizontal directions repeating themselves? Maybe I have misunderstood this, but it seems unnecessary. Maybe the authors could make this clearer by adding a few sentences to section 2.2.*

In the manuscript we have changed from circular to periodic boundary condition which is the more commonly used term. The periodic boundary condition does mean that here are infinitely many plumes in the horizontal directions (effectively the domain keeps on repeating itself in the horizontal). However, the plumes outside the domain where the camera is placed gets smaller and smaller, as seen by the camera, the further away they are from the camera. Careful selection of domain size and camera angles mostly avoid the problem of "ghost" plumes (cameras B, C, and D). However, for camera A, which views close to the horizon, some ghost were present as mentioned in the manuscript. We have re-written parts of section 2.2 to hopefully make this clearer.

**Response to comments from Referee #2**

**Comments**

1. *Fig 4 caption → indicate the results are for Camera A*
   Camera A mentioned in caption of Fig. 4.

2. *Page 10, line 10 → I believe that "Fig. 5" should be Fig. 4*
   Corrected.

3. *Page 18, line 10 → two instances of lower case "ssa" should be changed to upper case SSA.*
   Change made as suggested. In addition, ssa was also changed to SSA in Table 3.